# Influence of Zinc Oxide Nanoparticles and Char Forming Agent Polymer on Flame Retardancy of Intumescent Flame Retardant Coatings

**DOI:** 10.3390/nano10010042

**Published:** 2019-12-23

**Authors:** Tentu Nageswara Rao, T. Manohra Naidu, Min Soo Kim, Botsa Parvatamma, Y. Prashanthi, Bon Heun Koo

**Affiliations:** 1School of Materials Science and Engineering, Changwon National University, Changwon 51140, Gyeongnam, Korea; tnraochemistry@gmail.com (T.N.R.); alstn1991@changwon.ac.kr (M.S.K.); 2Department of Nuclear Physics, Andhra University, Visakhapatnam 530003, Andhra Pradesh, India; t.manoharanaidu@rediffmail.com; 3Department of Organic Chemistry, Gayathri P.G Courses, Gotlam, Vizianagaram, AP 535003, India; Parvathi1787@rediffmail.com; 4Department of Chemistry, Mahatma Gandhi University, Nalgonda 508254, Telangana, India; prashanthimgu@gmail.com

**Keywords:** IFR, char–foaming agent, ZnO NPs, PHRR, THR, TGA

## Abstract

Zinc oxide nanoparticles (ZnO NPs) were synthesized by a precipitation method, and a new charring–foaming agent (CFA) *N*-ethanolamine triazine-piperazine, melamine polymer (ETPMP) was synthesized via nucleophilic substitution reaction by using cyanuric chloride, ethanolamine, piperazine, and melamine as precursor molecules. FTIR and energy-dispersive X-ray spectroscopy (EDS) studies were employed to characterize and confirm the synthesized ETPMP structure. New intumescent flame retardant epoxy coating compositions were prepared by adding ammonium polyphosphate (APP), ETPMP, and ZnO NPs into an epoxy resin. APP and ETPMP were fixed in a 2:1 w/w ratio and used as an intumescent flame-retardant (IFR) system. ZnO NPs were loaded as a synergistic agent in different amounts into the IFR coating system. The synergistic effects of ZnO NPs on IFR coatings were systematically evaluated by limited oxygen index (LOI) tests, vertical burning tests (UL-94 V), TGA, cone calorimeter tests, and SEM. The obtained results revealed that a small amount of ZnO NPs significantly increased the LOI values of the IFR coating and these coatings had a V-0 ratings in UL-94 V tests. From the TGA data, it is clear that the addition of ZnO NPs could change the thermal degradation behaviors of coatings with increasing char residue percentage at high temperatures. Cone calorimeter data reported that ZnO NPs could decrease the combustion parameters including peak heat release rates (PHRRs), and total heat release (THR) rates. The SEM results showed that ZnO NPs could enhance the strength and the compactness of the intumescent char, which restricted the flow of heat and oxygen.

## 1. Introduction

Wood is one of the most sustainable, aesthetically pleasing and environmentally friendly materials. In recent years, demand for the use of wood and wood-based products for applications in both residential and nonresidential construction has increased. However, due to the inherent flammability of these products, they often contribute to unwanted fires, resulting in many injuries and fatalities. The use of wood is therefore limited by various safety requirements and regulations relating to its flammability and the propagation of fire characteristics. To overcome this problem and maintain fire safety, fire-resistant coatings must be developed.

Halogen-contained fire-resistant coatings are available on the market but cause a high level of pollution due to the release of toxic gases. Therefore, the use of halogen-free fire-retardant coatings is more recommended. In general, two types of halogen-free fire-retardant coatings, borate coatings and aluminum/magnesium coatings, are generally available. These coatings usually have high fire resistance, form a dense protective carbon and generate a low amount of smoke. However, the uses of these coatings have not shown a high level of practical application, as they are large and show incompatibility with binders. 

Currently, intumescent flame-retardant (IFR) coatings are widely used in applications of wood and steel coatings, because they do not release toxic gasses while burning. There are mainly three major components in the intumescent fire-retardant coating composition: (1) acid source, (2) charring agent, and (3) foaming agent [1,2]. While burning, intumescent components form a protective char with releasing of volatile gases such as ammonia and nitrogen [3]. The combination of APP as an acid source, pentaerythritol (PER) as a charring agent, and melamine (MEL) as a foaming agent forms a traditional IFR system. Some researchers reported that the use of this traditional IFR system did not produce good results and showed poor antioxidation and fire resistance behavior [4]. In addition to this, PER is a small molecule that is easily affected by moisture and results in poor performance of flame retardancy. 

Scientists have made great efforts to synthesize polymers that can exhibit both charring and foaming properties to solve this problem. In this path, triazine molecules were chosen to prepare charring–foaming agents (CFAs), as it contains a high amount of nitrogen and carbon units [5,6,7]. Additionally, some synthesized triazine derivatives showed good thermal stability [8,9,10,11]. However, it was reported that some of the synthesized triazine derived polymers showed lower thermal stability. Therefore, it is still a challenging task to synthesize highly thermal stable charring–foaming polymers. To increase the strength and the compactness of intumescent chars, some synergistic agents such as zeolite [12,13] and metal-based compounds [14,15,16] were added and investigated successfully. The researchers also reported that synergistic agents accelerated the intumescent mechanism and improved the compactness of chars. 

ZnO is a white-colored, inorganic and crystal compound. It is eco-friendly and has environmental and industrial catalytic applications. Due to high catalytic properties, ZnO can significantly replace the role of metal catalysts [17,18,19]. Based on catalytic applications of ZnO, we selected it as a synergist, and we expect that it can show positive effects on flame retardancy of IFR systems [20,21,22,23,24].

This study was performed to investigate the effects of ZnO on epoxy-based IFR coatings. Various amounts of ZnO were incorporated into an epoxy/IFR coating system. The effects of ZnO on epoxy/IFR coatings were investigated by limited oxygen index (LOI) test, vertical burning tests (UL-94 V), TGA, cone calorimeter tests, and SEM. 

## 2. Experimental

### 2.1. Materials

Bisphenol—an epoxy resin, Polyamide and APP (form II, particle size: 20 µm) was purchased from Samchun Pure Chemical Co., Ltd. (Pyungtaek, Gyeonggi-do, South Korea) The solvents of ethanolamine (analytical grade reagent; 99%), acetone (analytical grade reagent: 99.5%) and ethanol (analytical grade reagent; 98.4%) were procured from Sigma-Aldrich, Germany. Cyanuric chloride (99%), MEL (99.5%), piperazine (99.3%), zinc acetate (analytical regent; 99.1%), triethanolamine (TEA; 98.7%) and sodium hydroxide (NaOH 99.2%) pellets were obtained from Sigma-Aldrich, Korea South. All of the solvents and chemicals were used for experimental work as they received.

### 2.2. Preparation of Zinc Oxide Nanoparticles (NPs)

ZnO NPs preparation involved ultrasonic irradiation of a 50 mL aqueous solution containing 0.05 M of zinc acetate and 0.05 M of TEA for 2 h at 80 °C. The resultant precipitate was washed several times with deionized water followed by methanol, filtered and dried in air at 200 °C for 2 h and at 400 °C for 6 h [25,26,27,28].

### 2.3. Synthesis of N-Ethanolamine Triazine-Piperazine, Melamine Polymer (ETPMP)

We synthesized a CFA of ETPMP, and its synthetic path is shown in Scheme 1. This polymer was synthesized in three steps via nucleophilic substitution reaction. 

In the first step, 1 mol of cyanuric chloride and 1 L of acetone were poured in a clean and dry 3 L three-necked round-bottom flask, and the reaction was maintained below 10 °C. One mole of NaOH and 1 mol of ethanolamine were dissolved in distilled water, and this solution was added to the flask and stirred for 2 h. After the completion of the first step, 2-ethanolamine-4,6-dichloride-1,3,5-triazine was obtained as an intermediate, and it was not taken out from the three-necked round-bottom flask. In the second step, 0.5 mol of piperazine and 1 mol NaOH were dissolved in water, the mixed solution was added slowly into the flask for 2 h, and the temperature was increased to 45–50 °C under constant stirring for 4 h. In the final step of the synthesis, 0.3 mol of MEL and 1 mol of NaOH were completely dissolved in water, and this aqueous solution was added to the reaction flask. In this step, the reaction temperature was increased to 70 °C for the removal of acetone. This step was performed for 6 h to complete the reaction. Finally, the obtained white precipitate was collected by filtration and washed with ethanol three times to remove impurities from the targeted product of ETPMP.

### 2.4. Sample Preparation

IFR coatings were prepared by the mixing of APP and ETPMP (2:1 w/w) with ZnO (synergist) into the epoxy resin. APP, ETPMP, and ZnO NPs were mixed in the epoxy resin and poured into a dispersion mixer for homogeneous mixing. Then, the prepared coatings were brushed on plywood pieces for different characterizations. After coating, the plywood pieces were dried for two days in a ventilated room. The coating composition is listed in Table 1.

### 2.5. XRD Study

The XRD study was conducted by a D8 X-ray diffractometer model with Cu Kα radiation for the structural analysis of ZnO NPs. The scanning rate of 0.02° per second and a current of 30 mA over a 2θ range of 20°–80°.

### 2.6. FTIR Analysis

A Nicolet Aviator 360 FTIR spectrometer instrument (Nicolet, WI, USA) was used to record and monitor the FTIR spectra of the samples at a resolution of 2 cm^−1^. The FTIR spectra of the samples were monitored in a range of 4000–400 cm^−1^. 

### 2.7. TEM Analysis

The ZnO particle size was measured by TEM (TECNAI G2 TF20-ST, FEI, NCNST, China).

### 2.8. Energy-Dispersive X-Ray Spectroscopy (EDX) Analysis

EDX was used for the elemental composition of the ETPMP polymer and the formulation samples (Ep/IFR and Ep/IFR/ZnO-3%).

### 2.9. Flame Retardancy Test

For IFR samples, flame-retardant properties such as LOI and UL-94 V vertical burning tests were performed. The LOI test was performed at room temperature on an HC-2 C oxygen index instrument. This study was conducted by following the ASTM D2863–97 method guidelines. The coating formulations were applied on plywood sheets with a dimension of 130 mm X 6.5 mm X 3 mm to perform the LOI test.

In order to evaluate the UL-94 V ratings of the samples, a study was conducted on a CZF-2 instrument (Dongguan Lonroy Equipment co.,Ltd, Dongguan, China). The prepared coatings were applied on plywood pieces with a dimension of 130 mm X 6.5 mm X 3 mm and dried. The ASTM D3801 method guideline was followed, and ignition time, burning time, and flame retardation time for each sample were recorded. 

### 2.10. TGA Test

All sample thermograms were recorded on Perkin-Elmer equipment (TGA Q600, UK at a nitrogen gas flow rate of 20 mL/min. The temperature range was set from 30 to 800 °C with a heating rate of 10 °C/min. The weight of about 5–10 mg of each sample was taken for TGA analysis.

### 2.11. Combustion Test

Combustion behaviors, peak heat release rates (PHRRs), and total heat release (TRR) rates were determined by a microcalorimeter instrument (modal 11311; Federal Aviation Administration, Fire Testing Technology, UK). The test was conducted according to the ISO 5660 guidelines. For this study, 4–5 mg of each sample was put into a thin-walled, quartz capillary crucible, kept in a sample holder and heated at a constant rate of 260 K/min. 

### 2.12. SEM

The surface morphologies of ZnO NPs and char residues, which were obtained after performing TGA, were examined by SEM at a 3000 magnification. Before the SEM examination, a conductive gold layer was used for sputter coating of char residue surfaces. An accelerating voltage of 15 kV was applied for the examination. 

### 2.13. Strength of the Residual Charred Layer

An intumescent charred layer obtained after cone calorimeter analysis was used for strength analysis study. The intumescent charred layer was laid on a circular platform, and then a slightly increased pressure, at which the charred layer was damaged, was applied on the char layer. 

## 3. Results and Discussion

### 3.1. Characterization of ZnO NPs

SEM, TEM, XRD, and FTIR characterized the synthesized ZnO NPs, and the respective results are presented in Figure 1, Figure 2, Figure 3 and Figure 4. The SEM micrographs of ZnO NPs were observed at different magnifications. The high-resolution SEM images show the presence of NPs. The SEM images show the agglomeration of the particles with narrow particle size distributions [29,30,31]. The NPs average size was found to be about 20 nm. The size of the NPs was calculated using TEM and XRD patterns of ZnO nanopowders synthesized from zinc acetate. The XRD data were recorded using Cu Kα radiation. The intensity data were collected over a range of 20–80°. These XRD patterns show much sharper peaks. In Figure 3, distinctive ZnO peaks appear at 2θ of 31.51°, 34.658°, 36.453°, 47.74°, 56.78°, 62.99°, 66.38°, 68.06°, 69.16°, and 79.96° [32,33,34]. These patterns represent the wurtzite phase of ZnO. A definite line broadening of the diffraction peaks also resembles that the synthesized particles were in a dimension of nanometer range, and no characteristic peaks were observed other than ZnO peaks. The average crystalline size calculated by applying the Debye-Scherrer equation on the diffraction peaks was found to be 19.8 nm for zinc oxide particles synthesized from zinc acetate. The FTIR analysis was used to examine the chemical composition and the functional group of ZnO NPs in the range of 400–4000 cm^−1^ [35]. From Figure 4, various modes of vibration were observed in different regions of the FTIR spectrum. The absorption band at 3428 cm^−1^ was caused by the O–H stretching mode, which shows a small amount of water absorbed by the ZnO NPs. The band at 1624 cm^−1^ indicates the stretching mode of C=O. This functional group appears because of the incorrect decomposition of the acetate group in the precursor solution. The stretching modes of C=O and C–C were observed at 1393 and 1027 cm^−1^, respectively. For this sample, the absorption band at 443 cm^−1^ was attributed to the Zn–O stretching vibration mode.

### 3.2. FTIR Spectroscopy of ETPMP

The FTIR spectrum of the ETPMP polymer is shown in Figure 5, which contains a broad absorption peak at 3200–3450 cm^−1^ for the stretching vibrations of the N–H and O–H bonds. The peaks at 2936 and 2884 cm^−1^ are significantly attributed to the stretching vibration of the C–H bond in the –CH_2_-CH_2_– group. The peaks attributed at 1562, 1404, 1326 and 1054 cm^−1^ are for the ν_tr_, δ_O–H, νtr_-N and ν_C–O_, respectively. There is no peak for the stretching vibration of the C–Cl bond at 850 cm ^−1^, and it was demonstrated that all three chlorine groups of cyanuric chloride were substituted successfully. 

### 3.3. EDS Analysis of ETPMP

The elemental composition of ETPMP was studied by EDS, as shown in Figure 6. ETPMP exhibits three elemental peaks: one for the carbon element located at 0.3 keV, one for the oxygen element located at 0.42 keV, and one for the oxygen element located at 0.61 keV. 

### 3.4. Thermal Degradation Behavior of ETPMP

TGA analysis provides information about the thermal degradation of polymers. The thermal degradation behaviors of ETPMP are represented in Figure 7 and Table 2. These data showed that the prepared ETPMP was thermally more stable, because even at 800 °C it had 35.58% of char residues. Figure 7 represents that ETPMP showed thermal degradation in two major stages. Between 90 and 200 °C, the first stage of thermal degradation took place. In this stage, the mass loss of ETPMP of approximately 10% occurred due to the evaporation of solvents, water, and ammonia. In the case of the second degradation stage (280–560 °C), a great mass loss (around 45%) occurred. It may be due to the decomposition of a cross-linkage from ETPMP. As a result, large amounts of ammonia and nitrogen gasses were released. After the breakdown of the ETPMP molecular structure, a swollen char was produced along with foaming gases (ammonia and nitrogen). The TGA results of ETPMP clearly revealed that it has good charring and foaming properties at high temperature conditions.

### 3.5. FTIR Analysis of the Formulation Samples

The FTIR spectra were used to investigate bonding interactions as well as identifying functional groups in different formulation samples. The FTIR spectra of the Ep/IFR and Ep/IFR/ZnO-3% samples are shown in Figure 8. An absorption band at 443 cm^−1^ was observed in the Ep/IFR/ZnO-3% sample only. It was clearly indicated that ZnO was contained in this sample. In both samples, reaming bands (3269, 2944, 3883, and 1563 cm^−1^) were observed. The samples containing the groups OH, N–H, C–H, and C=O were based on these bands.

### 3.6. EDS Analysis of the Formulation Samples

The EDS spectra of the Ep/IFR and Ep/IFR/ZnO-3% formulations are shown in Figure 9. There are four significant peaks corresponding to carbon, nitrogen, oxygen, and phosphorus, respectively, according to the EDS spectra in Figure 9a. These results were compatible with the composition of the F0 formulation used for APP, the epoxy, and the CFA. There are five significant peaks corresponding to carbon, nitrogen, oxygen, phosphorus and zinc, respectively, in Figure 9b. These results were compatible with the composition of the F3 formulation used for ZnO NPs, APP, the epoxy, and the CFA. These results are in good agreement with previously reported results. 

### 3.7. Flame Retardancy Analysis

Table 3 shows the flame retardancy data of the samples. The data clearly showed that the pure epoxy resin with an LOI value of 21.2% was highly flammable. The addition of IFR (APP + ETPMP) to the pure epoxy resin greatly increased the LOI value up to 28.7%. This may result in APP being decomposed during the burning process to produce phosphoric acid, which then led to dehydration of ETPMP and ultimately produced a stable intumescent phosphorus carbonaceous char. At this stage, ETPMP also released noncombustible gases like ammonia and nitrogen, which made the phosphorus carbonaceous char swell. From the results, it can be observed that adding IFR (APP + ETPMP) and increasing ZnO _by_ up to 3% may increase LOI values. The addition of ZnO can cause the phosphor esterification or dehydration reaction to be fastened between APP and ETPMP. As APP burnt, it produced phosphoric acid, which was preferentially involved in dehydration reaction with ETPMP. From the structure of ETPMP, it is known to be a poly-hydroxyl triazine-based polymer derivative, so that it can easily be involved in dehydration reactions with APP. The UL-94 V test data of the samples showed that all of the samples, except the pure epoxy coating, had a V-0 rating. From LOI and UL-94 V results, it can be seen that the addition of ZnO may have a synergistic effect at an optimum loading concentration.

### 3.8. TGA Analysis

The thermal degradation patterns of the samples were performed by TGA under nitrogen atmosphere. The thermal degradation behaviors of the pure epoxy resin, Ep/IFR, Ep/IFR/ZnO-1%, Ep/IFR/ZnO-2%, and Ep/IFR/ZnO-3% are represented in Figure 10. The data showed that the pure Epoxy coating was highly flammable and it was completely burnt at above 600 °C. The pure epoxy exhibited initial thermal degradation at 250 °C. When the temperature was further increased, the interaction between the epoxy resin and an amine hardener was decomposed and finally led to the destruction of the resin. Compared to the pure epoxy coatings, the Ep/IFR coatings showed high thermal stability. They showed initial thermal degradation at 285 °C, and 20.87% of char residue was remained at 800 °C. At temperatures higher than 300 °C, the IFR system was involved in the intumescent mechanism. APP started to decompose and release mineral acids such as phosphoric acid and metaphosphoric acid at 300° C, which removed the water content from ETPMP via the esterification process. APP and ETPMP easily participated in phosphor esterification reaction, because ETPMP has an abundant amount of –OH groups. These –OH groups gently reacted with –OH groups of mineral acids, which were formed after the APP decomposition. This process finally led to forming a protective phosphorcarbonaceous char layer, which prevented oxygen flow and heat transfer. Compared to the Ep/IFR coatings, the Ep/IFR/ZnO-3% coatings showed a high percentage of char residues at 800 °C. Figure 10 clearly illustrates that the addition of ZnO could not affect the initial degradation temperature of the IFR system but greatly improved the char residues percentage from 20.87% to 34.36% for the Ep/IFR/ZnO-3% coatings. It may be due to the catalytic effect of ZnO on esterification reaction between APP and ETPMP, which can form the thermally insulating P–O–Zn–P bonds in the IFR system to strengthen the char. The effect of ZnO on the IFR system is represented in Scheme 2. 

### 3.9. Microcalorimeter Analysis

The combustion behaviors of the coatings have been evaluated by widely used microcalorimeters. Microcalorimeters provide information about combustion parameters of a material such as PHRRs, total heat release (THR) rates, and time to ignition (TTI), and these value are useful to predict the combustion behavior of a material in fire situations. 

Figure 11 and Table 4 show the PHRR data of the pure epoxy, Ep/IFR, Ep/IFR/ZnO-1%, Ep/IFR/ZnO-2%, and Ep/IFR/ZnO-3% coatings. From Figure 11, it can be seen that the pure epoxy coating was burned easily and the PHRR value is 696.12 W/g. The Ep/IFR, Ep/IFR/ZnO-1%, Ep/IFR/ZnO-2%, and Ep/IFR/ZnO-3% coatings had PHRR values of 398.11, 356.25, 317.57 and 287.62 W/g, respectively. 

TTI values were used to know the effect of a flame retardant on the ignition parameter. The data showed that the TTI values of the Ep/IFR, Ep/IFR/ZnO-1%, Ep/IFR/ZnO-2%, and Ep/IFR/ZnO-3% coatings are less than that of the pure epoxy resin. The TTI values for Ep/IFR, Ep/IFR/ZnO-1%, Ep/IFR/ZnO-2%, and Ep/IFR/ZnO-3% respectively, are 281, 278, 275, and 251 s lower than those of the pure epoxy resin (340 s).

The THR rate data of the pure epoxy, Ep/IFR, Ep/IFR/ZnO-1%, Ep/IFR/ZnO-2%, and Ep/IFR/ZnO-3% coatings are shown in Table 4. From the data, it can be clearly seen that the epoxy coating had the highest THR rate value of 113.38 kJ/g among the samples. In the case of the Ep/IFR, Ep/IFR/ZnO-1%, Ep/IFR/ZnO-2% and Ep/IFR/ZnO-3% coatings, the THR rate values are 73.14, 65.47, 56.76, and 41.71 kJ/g, respectively, which are remarkably decreased compared with that of the pure epoxy resin (112 kJ/g). During the combustion process, the IFR system decomposed gently and resulted in char formation. In addition to the loading of ZnO into the coating system, the intumescent mechanism occurred gently. This may be a possible reason for the decrease in TTI values. After the formation of a char, oxygen flow and heat transfer were not allowed, and hence the PHRR and THR rate values are decreased. Among all coatings, the Ep/IFR/ZnO-3% coating showed lower PHRR and THR rate values. It may be due to the fact that during the combustion process ZnO fastened the esterification reaction in the IFR system to form a protective char. ZnO was able to form cross-links with the IFR system to protect the char from crack and holes. ZnO could also make the char more compact, which obviously resulted in significant decreases of the PHRR and THR rate values. All these results concluded that the addition of ZnO promotes a well and good synergistic action to improve the combustion behaviors of the coatings. 

### 3.10. Morphology of the Char Residues

To know the relationship between the flame retardancy of coatings and the microstructures of intumescent chars, the char residues of Ep/IFR and Ep/IFR/ZnO-3% after cone calorimeter tests were imaged by SEM with a 3000 magnification. The SEM images of the charred layers of Ep/IFR and Ep/IFR/ZnO-3% are clearly shown in Figure 12. From Figure 12a, it is seen that the Ep/IFR char had more cracks with big holes, which explained the crumbliness and the fragility of the char residues. Figure 12b shows that the surface of the char residues of Ep/IFR/ZnO-3% was smoother, more homogenous and more compact than that of the Ep/IFR char. The holes and the cracks, which were formed in the Ep/IFR char, may be due to the release and the flow of volatile gases into a flame zone. The addition of ZnO can form cross-links with the IFR system. As a result, a compact and strengthened char was formed, which effectively prevented the flow of gases from the material to the flame zone. Therefore, there were no holes and cracks in the Ep/IFR/ZnO-3% char residue. These results concluded that the addition of ZnO at an optimum level can strengthen chars to increase the flame retardancy of coatings.

### 3.11. Strengths of the Residual Charred Layers

After the cone calorimeter test, the strengths of the intumescent char layers were measured with a compressing test machine. The strengths of the chars of Ep/IFR and Ep/IFR/ZnO-3% were 0.023 and 0.042 N, respectively. It means the Ep/IFR/ZnO-3% char layer was stronger than the Ep/IFR char layer, due to the fact that ZnO formed the cross-linkages with the IFR system, which made the char more compact and stronger. This strengthened char can effectively restrict the flow of heat transfer and oxygen. These results also confirmed that ZnO acts as a good synergist to improve char strength and toughness. 

## 4. Conclusions

A new triazine derivative ETPMP was synthesized via nucleophilic substitution reaction, and its structure was confirmed by FTIR and EDS analysis. Novel IFR–Epoxy–ZnO coating compositions were prepared. The addition of ZnO, even with a small quantity, into the Ep/IFR coating system can play a vital role in increasing LOI values and reaches UL-94 V ratings. The LOI and UL-94 V results revealed that the optimum amount of ZnO was 3%. The TGA data clearly indicated that the addition of ZnO can greatly increase the thermal stability of the coating by increasing the char residue percentage at high temperature [36,37]. APP and ETPMP can easily participate in phosphor esterification reaction at their decomposition temperatures, as ETPMP has a significant number of –OH groups. These –OH groups reacted gently to –OH groups of mineral acids formed after APP decomposition. This process finally resulted in the formation of a protective phosphorous carbonaceous char layer, which prevented the flow of oxygen and the transfer of heat. Microcalorimeter data confirmed that ZnO can effectively decrease heat-releasing properties. The SEM analysis and strength analysis data proved that the addition of ZnO at an optimum level can favor the formation of a homogenous and compact char layer during the burning of the coating materials.

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
