# Peer review of "Influence of Zinc Oxide Nanoparticles and Char Forming Agent Polymer on Flame Retardancy of Intumescent Flame Retardant Coatings"

_nanomaterials, 2019, doi:10.3390/nano10010042_

Round 1

Reviewer 1 Report

The paper report on a interesting property of ZnO micro-nanomaterials to be used in flame-retardant composites.

There are few major changes to be solved, but more in general although the scientific soundness of the paper, the text is very careless written, with several errors and mistakes. Please amend them.

Major comments:

-2.9 Adsorption isotherm measurements: what is the rational of using CO2 as adsorption gas and not the most conventional N2?

-Pores of 2.69 nm are detected from CO2 sorption measurements. What is the nature of these pores: interparticle porosity? Actually no mesoporous strucutre is visible from the TEM in Fig. 2, so a better explanation of the pore origin must be given

-To support the conclusion of the Char and polymer morpholgogy shown in Fig. 13, a FESEM imaging at the same magnification should be provided on the as-prepared materials and compared to those shown in Fig. 13

Minor comments:

-please increase the quality in terms of higher dpi of Scheme 1 and Scheme 2 (upper part)

-line 154: remove the bracket after Pyris Elmer ( )

-line 176: please add the verb in "The SEM micrographs of ZnO nanoparticles ARE or WERE observed at different magnifications"

- when saying "In the Fig.3 the distinctive ZnO peaks at 31.51, 34.658, 36.453, 183 47.74, 56.78, 62.99, 66.38, 68.06, 69.16 and 79.96 respectively [32-34]." the indexing of the peaks by Miller index should be mentioned (or refer to the graph in Fig.3), as well as a comparison with accredited data bank to confirm the wurtzite crystal habit

-figure 1 and 2 seems to be horizontally stretched, please rezsize them correctly

-Please correct: Fig. 4. spectrUM of ZnO NPs.

-Please correct: line 213, FTIR spectrum of the ETPMP polymer is shown in Fig.1. It should be Fig. 7, isn't it?

-Please correct: line 250:  it clearly indicates

and similarly line 266: The data clearly reveals

line 268: "It may be resulted due to that during" into "This fact is due to the fact that during"

-please reprhrase or conrrect the tense of the verb: Based on these bands the samples containing OH, N-251 H, C-H and C=O groups.

-line 169: to produce .... which then makeS involvING in...

-line 272: it can be observeD

-line 273: corrrect errors in "It may be due to that addition of ZnO"

-line 275: which is preferentially involveD

-line 277: so it can readily involve in ..

- line 299: The figure mentioned should be Fig. 11 not Fig. 4

-line 315: Fig 5 should be Fig.12

-line 343: Fig 6. From the Fig 6 (a) and (b) should refer to Fig. 13, please also uniform the stile of letters (not in red) as the other figures

Author Response

Reviewer-1

Respected Sir,

Thank you very much for your valuable comments and suggestions, accordingly we make corrections and updated in the revised manuscript.

There are few major changes to be solved, but more in general although the scientific soundness of the paper, the text is very careless written, with several errors and mistakes. Please amend them.

Major comments:

-2.9 Adsorption isotherm measurements: what is the rational of using CO2 as adsorption gas and not the most conventional N2?

Answer: Thank you very much for rising question sir, generally we have to measure CO2 and N2 adsorption by BET isotherm. In this case I have measured CO2 adsorption only because of CO2 gas is anti-flammable gas. We used ZnO NPs to make IFR coating formulations. For this reason I just measured CO2 adsorption, the ZnO adsorbed CO2 gas means it is good anti-flame filler. 

-Pores of 2.69 nm are detected from CO2 sorption measurements. What is the nature of these pores: interparticle porosity? Actually no mesoporous strucutre is visible from the TEM in Fig. 2, so a better explanation of the pore origin must be given.

Answer:Thank you for your question sir, generally materials are two types 1. Non pours materials and 2. Pours materials. For pours materials we have to measure pore size and pore volume by BET iso therm and we have to measure particle size by TEM also. Pore size measurement is physical phenomenon, here I am given example.

By using TEM and ImageJ software we measured average particle size.

A mesoporous material is a material containing pores with diameters between 2 and 50 nm, according to IUPAC nomenclature. For comparison, IUPAC defines microporous material as a material having pores smaller than 2 nm in diameter and microporous material as a material having pores larger than 50 nm in diameter.

As per my knowledge we cannot measure microporous material particle size by TEM. Mesoporous and microporous materials possible to do TEM analysis.

-To support the conclusion of the Char and polymer morpholgogy shown in Fig. 13, a FESEM imaging at the same magnification should be provided on the as-prepared materials and compared to those shown in Fig. 13.

Answer: Thank you for your good advice sir, accordingly I have maintained same magnification SEM figures in Fig 13 in revised manuscript.

 Minor comments:

-please increase the quality in terms of higher dpi of Scheme 1 and Scheme 2 (upper part)

Answer: Now I improved good resolutions (dpi) of Scheme 1 and Scheme 2 in revised manuscript.

-line 154: remove the bracket after Pyris Elmer ( )

Answer: Removed bracket

-line 176: please add the verb in "The SEM micrographs of ZnO nanoparticles ARE or WERE observed at different magnifications"

Answer: magnification value included

- when saying "In the Fig.3 the distinctive ZnO peaks at 31.51, 34.658, 36.453, 183 47.74, 56.78, 62.99, 66.38, 68.06, 69.16 and 79.96 respectively [32-34]." the indexing of the peaks by Miller index should be mentioned (or refer to the graph in Fig.3), as well as a comparison with accredited data bank to confirm the wurtzite crystal habit.

Answer: Thank for your good observation sir, I have mentioned wurtzite phase of ZnO

-figure 1 and 2 seems to be horizontally stretched, please rezsize them correctly

Answer: Corrected

-Please correct: Fig. 4. Spectr FTIR of ZnO NPs.

Answer: Corrected

-Please correct: line 213, FTIR spectrum of the ETPMP polymer is shown in Fig.1. It should be Fig. 7, isn't it?

Answer: Yes sir, now I have corrected

-Please correct: line 250:  it clearly indicates

Answer: The sentence has been reframed

and similarly line 266: The data clearly reveals

Answer: The sentence has been reframed

line 268: "It may be resulted due to that during" into "This fact is due to the fact that during"

Answer: The sentence has been reframed

-please reprhrase or conrrect the tense of the verb: Based on these bands the samples containing OH, N-251 H, C-H and C=O groups.

Answer: The sentence has been reframed

-line 169: to produce .... which then makeS involvING in...

Answer: The sentence has been reframed

-line 272: it can be observed

Answer: The sentence has been corrected

-line 273: correct errors in "It may be due to that addition of ZnO"

Answer: The sentence has been corrected

-line 275: which is preferentially involved

Answer: The sentence has been corrected

-line 277: so it can readily involve in ..

Answer: The sentence has been reframed

- line 299: The figure mentioned should be Fig. 11 not Fig. 4

Answer: I have been corrected

-line 315: Fig 5 should be Fig.12

Answer: I have been corrected

-line 343: Fig 6. From the Fig 6 (a) and (b) should refer to Fig. 13, please also uniform the stile of letters (not in red) as the other figures.

Answer: I have been corrected

Reviewer 2 Report

Proofread is required since some sentences, such as on line 19, line 34, line 42, and many more, are not properly constructed. There are also typos, such as on line 145, line 150, and many more. Please correct them. Figure numbers are wrongfully assigned, and the figure numbers in the manuscript do not match all the figures. Please check. The aim of this research should match the title about the influence of zinc oxide nanoparticles on flame retardancy. However, the paragraphs covering the aim in the manuscript seem not sufficient enough to reveal the role of ZnO in IFR. The role of ZnO in IFR discussed in the manuscript seems to be known knowledge instead of a research results coming out from this study. Please rearrange your manuscript, possibly by omitting surplus characterization portions, so that the research aim can be focused.

Author Response

Reviewer-2

Respected Sir,

Thank you very much for your valuable comments and suggestions, accordingly we make corrections and updated in the revised manuscript.

Proofread is required since some sentences, such as on line 19, line 34, line 42, and many more, are not properly constructed. There are also typos, such as on line 145, line 150, and many more. Please correct them.

  Answer: Thank you very much for your wrong english sentence formations, according to your suggestions I have corrected in revised manuscript.

Figure numbers are wrongfully assigned, and the figure numbers in the manuscript do not match all the figures. Please check.

Answer:  Thank you very much for your observation, accordingly I have corrected figure numbers in revised manuscript.

The aim of this research should match the title about the influence of zinc oxide nanoparticles on flame retardancy. However, the paragraphs covering the aim in the manuscript seem not sufficient enough to reveal the role of ZnO in IFR. The role of ZnO in IFR discussed in the manuscript seems to be known knowledge instead of a research results coming out from this study. Please rearrange your manuscript, possibly by omitting surplus characterization portions, so that the research aim can be focused. 

Answer: Thank you for your advice sir. In addition, in this study I changed the composition of ZnO NPs, thermal and flame retardancy properties have changed.  The addition of ZnO can significantly increase the thermal stability of the coating by increasing the percentage of char residues at high temperatures. I mentioned the import role of ZnO NPs in the conclusion section. Furthermore, I have characterized ZnO NPs such as CO2 adsorption, surface area and pore size measurement that CO2 adsorption and surface area are important information for intumescent flame retardant coatings.

Round 2

Reviewer 1 Report

The authors have replied to most of the concerns raised by the reviewer, however they have not given any precise answer to the following question:

-Pores of 2.69 nm are detected from CO2 sorption measurements. What is the nature of these pores: interparticle porosity? Actually no mesoporous strucutre is visible from the TEM in Fig. 2, so a better explanation of the pore origin must be given. (Authors have answered just (in contrast they just describing the techique and ho pores are classified, which is very clear to me, but did not answered about their sample). I strongly believe that the porosity they observe by CO2 soprtion measuremetn is due to inter-particle porosoty and the nanoparticles  themselves are not porous at all. I suggest thus author to make ths clear in the paper.

Still a minor change, to imrove the figure can be done:

-Figure 8 is too dark and with a blue cackgroud different form the further figures 11. Please reder homogeneous the graph presentation. Funthermore the scale on the x axis of Fig 8 is too broad, and a scale resizing from 0 to 5 keV can be used to better enlarge the EDX peaks.

Author Response

Respected Sir,

Thank you so much for your comments and valuable suggestions.

The authors have replied to most of the concerns raised by the reviewer, however they have not given any precise answer to the following question:

-Pores of 2.69 nm are detected from CO2 sorption measurements. What is the nature of these pores: interparticle porosity? Actually no mesoporous strucutre is visible from the TEM in Fig. 2, so a better explanation of the pore origin must be given. (Authors have answered just (in contrast they just describing the techique and ho pores are classified, which is very clear to me, but did not answered about their sample). I strongly believe that the porosity they observe by CO2 soprtion measuremet is due to inter-particle porosoty and the nanoparticles themselves are not porous at all. I suggest thus author to make ths clear in the paper.

Answer: Thank you very much for your good observations and very good technical information. I am sorry to say that, in fact, my subordinate (co-author) inadvertently named Chitosan dopped ZnO NPs CO2 sorption and pore size details were mentioned in the manuscript. The details belonged to the Chitosan doped ZnO NPs. The details of CO2 adsorption, surface area and pore size measurement of ZnO NPs are now deleted.

 Still a minor change, to imrove the figure can be done:

-Figure 8 is too dark and with a blue cackgroud different form the further figures 11. Please reder homogeneous the graph presentation. Funthermore the scale on the x axis of Fig 8 is too broad, and a scale resizing from 0 to 5 keV can be used to better enlarge the EDX peaks.

Answer: According to your suggestion, I have modified the Figure 8 and also the scale of X axis 0 to 5 keV.

Reviewer 2 Report

The content of the study is too divergent, so that the focus cannot be highlighted. Too many paragraphs on synthesis and characterization, where those synthesis and characterization do not directly lead to the main focus of the influence of ZnO and CFA on the flame retardancy of IFR coating. Too many acronyms make the article difficult to read. Especially for those on line 352, PHRR, THR, and MLR. The readability could be improved if those “acronyms” can be spell out in the subsequent paragraphs. On line 367, TTI is not specified. Please check. Figure numbers are wrongfully assigned, for example on line 338, Fig.4. Please check. The mechanism of IFR shown in Scheme 2 in page 14 of the manuscript does not seem to conclude from the study illustrated before Scheme 2. Scheme 2 seems to be a hypothesis by authors and is not derived from the preceding paragraphs. More details should be provided to link the mechanism with the preceding study to justify the focus of the manuscript.

Author Response

Respected Sir,

Thank you so much for your comments and valuable suggestions.

The content of the study is too divergent, so that the focus cannot be highlighted. Too many paragraphs on synthesis and characterization, where those synthesis and characterization do not directly lead to the main focus of the influence of ZnO and CFA on the flame retardancy of IFR coating. Too many acronyms make the article difficult to read. Especially for those on line 352, PHRR, THR, and MLR. The readability could be improved if those “acronyms” can be spell out in the subsequent paragraphs. On line 367, TTI is not specified. Please check. Figure numbers are wrongfully assigned, for example on line 338, Fig.4. Please check. The mechanism of IFR shown in Scheme 2 in page 14 of the manuscript does not seem to conclude from the study illustrated before Scheme 2. Scheme 2 seems to be a hypothesis by authors and is not derived from the preceding paragraphs. More details should be provided to link the mechanism with the preceding study to justify the focus of the manuscript. 

Answer: Line Number 352, improved if those acronyms of PHRR, THR and TTI. Line 367 and Table 4, TTI values are mentioned. Line No. 338, Fig.4 was corrected. According to Scheme 2 I have concluded in the Thermogravimetric analysis section and conclusion. Micro calorimeter analysis and thermogravimetric analysis demonstrated the significance of ZnO and CFA and how to improve combustion and thermal properties with the influence of ZnO and CFA.
